# Numerical Modeling of Flexible Net Panels under Steady Flow Using a Coupled Fluid–Structure Partitioned Scheme

**Lingyun Xu [1], Hongde Qin [1], Peng Li [1,2,\*] and Zhijing Xu [3]**

1 College of Shipbuilding Engineering, Harbin Engineering University, Harbin 150001, China; lingyun.xu@foxmail.com (L.X.); qinhongde@hrbeu.edu.cn (H.Q.)
2 Yantai Research Institute and Graduate School, Harbin Engineering University, Yantai 264006, China
3 Ningbo Institute, Dalian University of Technology, Ningbo 315016, China; xuzj_nbi@dlut.edu.cn
\* Correspondence: peng.li@hrbeu.edu.cn

**Abstract:** Fluid–structure interactions of flexible net panels are complex and lack sufficient exploration. To examine the flow characteristics of flexible net panels with large deformation, we propose a partitioned coupling scheme in this paper. The coupled fluid–structure equations are solved separately under finite volume and finite element frameworks. The interface traction from the fluid solver is considered as a Neumann boundary condition for the solid domain, and the interface velocity is applied as a Dirichlet boundary condition for the fluid problem. Then, the forces can be transferred along the interface via Dirichlet-to-Neumann mapping. The results show that both the drag coefficient and the velocity reduction increase alongside the net solidity ratio ($Sn$), but they decrease as the Reynolds number/attack angle increases. A comparative study of drag coefficients is made between the present numerical simulations and the analytical predictions. This paper also examines the velocity distribution and vortex formation of flexible net panels. A single vortex forms in the shear layers and the wake when $Sn = 0.16$, and a pair of vortices mostly forms in the wake when $Sn = 0.33$. The vertical net twines predominantly affect the formation of the vortex behind the net, leading to delayed vortex shedding. The flow exhibits wake interactions due to the interference between the net twines in the high-solidity net panel. No such interference occurs in the low-solidity net panel, but the altered shear layers could cause severe velocity fluctuations in the near field.

**Keywords:** flexible net panel; fluid–structure interactions; partitioned coupling scheme; vorticity

## 1. Introduction

Large-scale offshore aquaculture farming has become a sustainable method of feeding the world, since the practice of farming the ocean is growing rapidly. However, potentially high-energy oceanic conditions such as strong currents can cause net deformation, significantly influencing the current forces on and oxygen levels inside the cages. Understanding the fluid–structure interactions supports the longevity of the systems and is, therefore, vitally important for designing offshore aquaculture cages.

Research, including experimental studies and analytical/numerical predictions for estimating the fluid forces on nets, have been examined thoroughly [1]. According to their report, to numerically analyze the nets in currents and waves, a structure solution method, such as the finite element method [2–8]; a fluid solution method, such as computational fluid dynamics (CFD) method [9–14]; and a coupled fluid–structure interactions (FSI) method [15–18] have been established. Alongside these three methods, two hydrodynamic force models are suggested to calculate viscous forces on nets [19]. Dependent on how a net is approximated, the hydrodynamic force models can either be the Morison-type force model or the screen-type force model.

The finite element method (FEM) focuses on the structural models. The net is modeled with triangular/truss elements or mass-spring systems. The hydrodynamic forces on

the net are calculated based on the Morison equation or the screen-type force model. The Newton–Raphson and Rouge–Kutta approaches are used to solve the equations of motion. These FEM models are computationally efficient because they model the net in a mesh grouping manner without losing the geometrical and physical properties. However, a comparative study showed that FEM-based models over-predicted the drag force of the net [20]. One possible reason is that the interference of the flow field is not considered. Thus, further development of the numerical models is suggested.

CFD-based numerical models such as the porous media model have been developed to examine the flow field and velocity distribution around and inside net cages. In the porous model, the net is approximated by a porous media sheet instead of an equivalent system of cylinders. Considering the fluid viscosity, the Reynolds-averaged Navier–Stokes (RANS) equations and turbulence models are used to ascertain the underlying fluid mechanisms. However, the porous media model is based on the Darcy–Forchheimer law and it requires the prescribed resistance coefficients. Chen and Christensen [16] directly used free coefficients, namely the normal and tangential resistance coefficients, in porous media simulations by neglecting the Darcy–Forchheimer dependencies, since it was challenging to incorporate all aspects of the properties influencing the force on a net. Note that using a porous media model leads to a constant pressure-loss zone rather than a natural pressure drop. Moreover, the strategy used to resolve net deformation in the porous media model is likely the overset mesh, which leads to overlapped regions in the intersectional zones [16,21]. These issues need to be addressed.

FEM offers effective structural models, while a CFD-based porous media model facilitates the fluid field-analysis. Together, the structural model (e.g., lumped-mass model) and the porous media model have been used to examine the FSI between the fluid and the flexible nets [15]. This coupled FSI model takes net deformation into account using an iterative scheme. It assumes that a steady condition is satisfied at each iteration. Prompted by Bi et al. [15], Chen and Christensen [16] conducted an FSI analysis of an aquaculture net cage under steady flow using a coupled model, which combines the porous media model and the lumped-mass structural model. However, it seems that their model was unable to predict net twines' flow characteristics. Most recently, Martin et al. [21] developed a Lagrangian approach for the coupled simulation of aquaculture net panels in a Eulerian fluid model based on the immersed boundary methods. Instead of using a porous media model, the net was represented by Lagrangian points, and the RANS equations were solved in a Eulerian fluid domain. Their model has been validated for rigid [18] and flexible nets [22]. Still, the flow characteristics of the individual net twines and their interactions have not been settled. These drawbacks of the porous media model and the net deformation in the FSI framework should be resolved to significantly advance the numerical modeling of flexible nets and cages.

Depending on whether the governing fluid–structure equations are solved together or separately, the numerical schemes of the FSI problems are classified into two approaches: monolithic and partitioned schemes. In a monolithic scheme [23–27], the coupled nonlinear governing equations are solved together using an iterative technique such as the Newton–Raphson scheme. This approach requires coupling matrices and much more computational resources. In the partitioned/staggered scheme [28–30], the coupled equations are solved separately in the fluid and solid subdomains to obtain the fluid–structure solutions. This approach allows variables to be transferred between individual solvers via an interface [31–34]. It seems that the partitioned approaches are virtually favored for accuracy and efficiency as they support staggered coupling among fluid, structural and mesh movement in either a loose or tight manner.

To facilitate a better understanding of the flow characteristics of the flexible net panel and mutual interactions of the net twines, this paper proposes a coupled fluid–structure partitioned scheme, to conduct accurate and computationally efficient simulations for the FSI problems regarding net deformation under steady flow. This partitioned coupling scheme has second-order accuracy in time. By constructing the subdomains for the fluid

and the structure, and imposing Dirichlet boundary conditions on the fluid and Neumann boundary conditions on the structure, the governing fluid–structure equations can be solved separately. Then the variables, namely the forces, are transferred along the interface via a predictor–corrector scheme. An arbitrary Lagrangian–Eulerian formulation is employed to accommodate the independent mesh motion at each time step.

In what follows, we first introduce the governing fluid–structure equations in Section 2. This is followed by the numerical solutions of fluid–structure equations in Section 3. Section 4 presents the partitioned coupling formulation. Section 5 elaborates on the numerical setup and verification. With this, we then provide the velocity and vorticity distribution of flexible net panels under steady flow in Section 6. Section 7 gives a summary of the method and the conclusions of this paper.

## 2. Governing Fluid–Structure Equations

### 2.1. Governing Equations for the Fluid

We consider a three-dimensional flexible net panel $\Omega^s$ interacting with an incompressible steady flow $\Omega^f(t)$. The Navier–Stokes equations for an isothermal, incompressible, viscous fluid flow on a deformable spatial domain $\Omega^f(t)$ in an arbitrary Lagrangian–Eulerian (ALE) formulation are:

$$\rho^f \frac{\partial \boldsymbol{u}^f}{\partial t} + \rho^f (\boldsymbol{u}^f - \boldsymbol{w}) \cdot \nabla \boldsymbol{u}^f = \nabla \cdot \boldsymbol{\sigma}^f + \boldsymbol{f}^f \text{ in } \Omega^f(t) \tag{1}$$

$$\nabla \cdot \boldsymbol{u}^f = 0 \text{ in } \Omega^f(t) \tag{2}$$

where $\rho^f$ is the fluid density; $\boldsymbol{u}^f = \boldsymbol{u}^f(\boldsymbol{x}, t)$ and $\boldsymbol{w} = \boldsymbol{w}(\boldsymbol{x}, t)$ represent the fluid and mesh velocities at each defined spatial point $\boldsymbol{x} \in \Omega^f(t)$, respectively; $\boldsymbol{f}^f$ is the fluid body force; and $\boldsymbol{\sigma}^f$ is the Cauchy–stress tensor for a Newtonian fluid, written as:

$$\boldsymbol{\sigma}^f = -p\boldsymbol{I} + 2\mu^f \boldsymbol{\epsilon}^f \left( \boldsymbol{u}^f \right), \ \boldsymbol{\epsilon}^f \left( \boldsymbol{u}^f \right) = \frac{1}{2} \left[ \nabla \boldsymbol{u}^f + \left( \nabla \boldsymbol{u}^f \right)^T \right] \tag{3}$$

where $p$ is the fluid pressure; $\boldsymbol{I}$ denotes the second-order identity tensor; $\mu^f$ is the dynamic viscosity of the fluid; $\boldsymbol{\epsilon}^f$ is the rate-of-strain tensor; and superscript $T$ means transpose.

The ALE formulated Navier–Stokes Equations (1) and (2) in the weak form can be written as:

$$\begin{aligned}
\int_{\Omega^f(t)} \rho^f \left( \frac{\partial \boldsymbol{u}^f}{\partial t} + \left( \boldsymbol{u}^f - \boldsymbol{w} \right) \cdot \nabla \boldsymbol{u}^f \right) \cdot \boldsymbol{\phi}^f d\Omega + \int_{\Omega^f(t)} \boldsymbol{\sigma}^f : \nabla \boldsymbol{\phi}^f d\Omega \\
= \int_{\Omega^f(t)} \boldsymbol{f}^f \cdot \boldsymbol{\phi}^f d\Omega + \int_{\Gamma_n^f(t)} \boldsymbol{T}^f \cdot \boldsymbol{\phi}^f d\Gamma + \int_{\Gamma(t)} \left( \boldsymbol{\sigma}^f(\boldsymbol{x}, t) \cdot \boldsymbol{n}^f \right) \cdot \boldsymbol{\phi}^f(x) d\Gamma
\end{aligned} \tag{4}$$

$$\int_{\Omega^f(t)} q \nabla \cdot \boldsymbol{u}^f d\Omega = 0 \tag{5}$$

Here, $\boldsymbol{\phi}^f$ and $q$ are the smooth test functions for the fluid velocity and pressure, respectively; $\Gamma_n^f(t)$ represents the fluid Neumann boundary, where $\boldsymbol{\sigma}^f(\boldsymbol{x}, t) \cdot \boldsymbol{n}^f = \boldsymbol{T}^f$; $\boldsymbol{n}^f$ is the normal to the fluid boundary. The update to the deformable fluid subdomain is enforced by the ALE formulation [35,36].

The Reynolds-averaged equation is not closed, so it must be solved in combination with the turbulence model equation. In this paper, the shear stress transmission $k - \omega$ turbulence model (SST $k - \omega$) is selected, and the expression is as follows:

$$\rho \frac{\partial k}{\partial t} + \rho \frac{\partial}{\partial x_i}(k u_i) = \frac{\partial}{\partial x_j} \left( \Gamma_k \frac{\partial k}{\partial x_j} \right) + G_k - Y_k + S_k \tag{6}$$

$$\rho \frac{\partial \omega}{\partial t} + \rho \frac{\partial}{\partial x_i}(\omega u_i) = \frac{\partial}{\partial x_j} (\Gamma_\omega \frac{\partial \omega}{\partial x_j}) + G_\omega - Y_\omega + D_\omega + S_\omega \tag{7}$$

Here, $k$ is the turbulent kinetic energy; $\omega$ is the turbulent dissipation rate; $G_k$ and $G_\omega$ are the generation terms of turbulent kinetic energy; and $\Gamma_k$ and $\Gamma_\omega$ represent the effective diffusivity of $k$ and $\omega$. $Y_k$ and $Y_\omega$ represent the dissipation terms of $k$ and $\omega$, respectively; $D_\omega$ is the cross-diffusion term; and $S_k$ and $S_\omega$ are the user-defined source terms.

### 2.2. Governing Equations for the Structure

The net panel subjected to waves and currents can be modeled as either equivalent truss elements [4,6] or mass-spring systems [37–39]. The deformation of the net panel is governed by the structural equation:

$$\rho^s \frac{\partial \boldsymbol{u}^s}{\partial t} = \nabla \cdot \boldsymbol{\sigma}^s + \boldsymbol{f}^s \text{ in } \Omega^s \tag{8}$$

where $\rho^s$ is the structure density; $\boldsymbol{u}^s = \boldsymbol{u}^s(\boldsymbol{x}, t)$ is the velocity of the net panel defined at each material point $\boldsymbol{x} \in \Omega^s$; $\boldsymbol{f}^s$ represents the external fluid forces acting on the net panel; and $\boldsymbol{\sigma}^s$ denotes the first Piola–Kirchhoff stress tensor. The solution technique of these differential equations is dependent on Newmark-β method (Newmark, 1959).

In the present study, we use the principle of virtual work to express the equations of motion and equilibrium of stresses acting on the net panel. Consider a net panel with a mass density $\rho^s$ that deforms under external fluid forces, where in representative points on the net panel are specified by their position vectors. The weak form of the structural dynamics Equation (8) can be written as:

$$\int_{\Omega^s} \rho^s \frac{\partial \boldsymbol{u}^s}{\partial t} \cdot \boldsymbol{\phi}^s d\Omega + \int_{\Omega^s} \boldsymbol{\sigma}^s : \nabla \boldsymbol{\phi}^s d\Omega$$
$$= \int_{\Omega^s} \boldsymbol{f}^s \cdot \boldsymbol{\phi}^s d\Omega + \int_{\Gamma_n^s} \boldsymbol{T}^s \cdot \boldsymbol{\phi}^s d\Gamma + \int_{\Gamma_0} (\boldsymbol{\sigma}^s(\boldsymbol{x}, t) \cdot \boldsymbol{n}^s) \cdot \boldsymbol{\phi}^s(x) d\Gamma \tag{9}$$

Here, $\boldsymbol{\phi}^s$ is the displacement function that maps each Lagrangian point $\boldsymbol{x} \in \Omega^s$ to its deformed position at time $t$; $\Gamma_n^s$ represents the solid Neumann boundary and $\boldsymbol{\sigma}^s(\boldsymbol{x}, t) \cdot \boldsymbol{n}^s = \boldsymbol{T}^s$; and $\Gamma_0$ denotes the fluid–structure interface $\Gamma(t)$ at time $t = 0$.

### 2.3. Governing Equations for the Fluid–Structure Interface

We only consider the external fluid forces as the solid Neumann boundary. Two interface boundary conditions corresponding to the traction and velocity continuity conditions must be satisfied along the fluid–structure interface. The weak form of the traction continuity condition is given as:

$$\int_{\Gamma(t)} \left( \boldsymbol{\sigma}^f(\boldsymbol{x}, t) \cdot \boldsymbol{n}^f \right) \cdot \boldsymbol{\phi}^f(x) d\Gamma + \int_{\Gamma_0} (\boldsymbol{\sigma}^s(\boldsymbol{x}, t) \cdot \boldsymbol{n}^s) \cdot \boldsymbol{\phi}^s(x) d\Gamma = 0 \tag{10}$$

One may observe that $\boldsymbol{u}^f$ and $\boldsymbol{u}^s$ are defined on different domains $\Omega^f(t)$ and $\Omega^s$, respectively, and the condition

$$\boldsymbol{\phi}^f(\boldsymbol{\phi}^s(x, t)) = \boldsymbol{\phi}^s(x) \tag{11}$$

Can be achieved by considering a conforming mesh along the interface $\Gamma_0$ with $\boldsymbol{\phi}^s(x, t)$ being the position vector of the deformable net panel.

## 3. Numerical Solutions of Fluid–Structure Equations

### 3.1. Numerical Solutions of the Fluid Equations

The ALE formulations of the fluid equations can be written in dimensionless form as:

$$\frac{\partial \boldsymbol{u}}{\partial t} + (\boldsymbol{u} - \boldsymbol{w}) \cdot \nabla \boldsymbol{u} + \nabla p = \frac{1}{Re} \nabla^2 \boldsymbol{u} + \boldsymbol{f}^f \tag{12}$$

$$\nabla \cdot \boldsymbol{u} = 0 \tag{13}$$

We use a finite volume method to discretize the fluid equations. The pressure-implicit splitting of operators (PISO) scheme is employed for the velocity-pressure coupling. The

spatial discretization of the pressure and momentum is conducted using second-order and second-order upwind schemes, respectively. The temporal discretization is conducted using a second-order implicit scheme.

By introducing the intermediate velocity, $u^*$, the velocity and pressure fields are calculated separately to avoid pressure oscillation. The solution procedure of the fluid equations is based on Chorin [40] for pressure matching and is described as:

Step 1: Predict the velocity:

$$\frac{u^* - u^n}{\Delta t} + (u^n - w^n) \cdot \nabla u^n = \frac{1}{Re} \nabla^2 u^n + \frac{\Delta t}{2} (u^n - w^n) \cdot \nabla ((u^n - w^n) \cdot \nabla u^n) \quad (14)$$

Step 2: Update the pressure:

$$\nabla^2 p^{n+1} = \frac{\nabla \cdot u^*}{\Delta t} \quad (15)$$

Step 3: Correct the velocity:

$$\frac{u^{n+1} - u^*}{\Delta t} = -\nabla p^{n+1} + \frac{\Delta t}{2} (u^n - w^n) \cdot \nabla p^n \quad (16)$$

where $Re$ denotes the Reynolds number; $\Delta t = t^{n+1} - t^n$ is the time length between steps $[t^n, t^{n+1}]$. The inlet velocity is prescribed for Steps 1 and 3; the outlet pressure is set for Step 2; and the algebraic equations are solved by the Algebraic Multigrid (AMG) solver.

### 3.2. Numerical Solutions of the Structural Dynamic Equations

The generalized-$\alpha$ method [41] is employed for the nonlinear structural dynamics. The virtual work equation for the structural dynamics can be written as the following system of equations:

$$M^S \ddot{d}^{n+1-\alpha_m^s} + C^S \dot{d}^{n+1-\alpha_f^s} + K^S d^{n+1-\alpha_f^s} = \mathcal{R}_s^{n+1-\alpha_f^s} \quad (17)$$

where $M^S$ is the mass matrix; $C^S$ is the damping matrix; $K^S$ is the stiffness matrix; and $\mathcal{R}_s$ is the force vector. Based on the generalized-$\alpha$ method [41], these vectors are expressed by:

$$\ddot{d}^{n+1-\alpha_m^s} = (1 - \alpha_m^s) \ddot{d}^{n+1} + \alpha_m^s \ddot{d}^n \quad (18)$$

$$\dot{d}^{n+1-\alpha_f^s} = (1 - \alpha_f^s) \dot{d}^{n+1} + \alpha_f^s \dot{d}^n \quad (19)$$

$$d^{n+1-\alpha_f^s} = (1 - \alpha_f^s) d^{n+1} + \alpha_f^s d^n \quad (20)$$

$$\mathcal{R}_s^{n+1-\alpha_f^s} = (1 - \alpha_f^s) \mathcal{R}_s^{n+1} + \alpha_f^s \mathcal{R}_s^n \quad (21)$$

We consider the implicit Newmark time integration method [42] for the acceleration and velocity at $t^{n+1}$:

$$\ddot{d}^{n+1} = \frac{1}{\beta \Delta t^2} (d^{n+1} - d^n) - \frac{1}{\beta \Delta t} \dot{d}^n - \frac{1 - 2\beta}{2\beta} \ddot{d}^n \quad (22)$$

$$\dot{d}^{n+1} = \frac{\gamma}{\beta \Delta t} (d^{n+1} - d^n) - \frac{\gamma - \beta}{\beta} \dot{d}^n - \frac{\gamma - 2\beta}{2\beta} \Delta t \ddot{d}^n \quad (23)$$

The acceleration and velocity vectors at the generalized midpoints are:

$$\ddot{d}^{n+1-\alpha_m^s} = \frac{1 - \alpha_m^s}{\beta \Delta t^2} (d^{n+1} - d^n) - \frac{1 - \alpha_m^s}{\beta \Delta t} \dot{d}^n - \frac{1 - \alpha_m^s - 2\beta}{2\beta} \ddot{d}^n \quad (24)$$

$$\dot{d}^{n+1-\alpha_f^s} = \frac{(1 - \alpha_f^s)\gamma}{\beta \Delta t} (d^{n+1} - d^n) - \frac{(1 - \alpha_f^s)\gamma - \beta}{\beta} \dot{d}^n - \frac{(1 - \alpha_f^s)(\gamma - 2\beta)}{2\beta} \Delta t \ddot{d}^n \quad (25)$$

The time integration parameters $\beta$, $\gamma$, $\alpha_m^s$ and $\alpha_f^s$ are expressed as:

$$\beta = \frac{1}{4}(1 - \alpha_m^s + \alpha_f^s)^2, \ \gamma = \frac{1}{2} - \alpha_m^s + \alpha_f^s, \ \alpha_m^s = \frac{2\rho_\infty^s - 1}{\rho_\infty^s + 1}, \ \alpha_f^s = \frac{\rho_\infty^s}{\rho_\infty^s + 1} \tag{26}$$

where the spectral radius $\rho_\infty \in [0,1]$. The fluid and solid subdomains are solved iteratively for partitioned treatment and domain decomposition. The next section of this paper will address the coupled fluid–structure matrix formulation and iterative force correction procedure.

## 4. Partitioned Strong Coupling Formulation

The structural dynamic system regarding fluid–structure interaction discretized by a finite element method gives coupled linear equations. The system of Equations (17)–(26) includes 12 equations and 12 unknown terms concerning displacement, velocity, acceleration and force. All quantities associated with time instant $t^n$ are known. Eliminating all quantities associated with fractional time instants between $t^n$ and $t^{n+1}$, together with the Dirichlet-to-Neumann mapping along the interface

$$d^f = d^s = d^f, \ \dot{d}^f = \dot{d}^s, \ \mathcal{R} = \mathcal{R}_s = -\mathcal{R}_f \tag{27}$$

yields

$$\left\{ \begin{matrix} d^{n+1} \\ \dot{d}^{n+1}\Delta t \\ \ddot{d}_s^{n+1}\Delta t^2 \\ d_f^{n+1}\Delta t^2 \\ \mathcal{R}^{n+1}\Delta t^2 \end{matrix} \right\} = \left[ \quad \mathcal{A}_{\text{strong}} \quad \right] \left\{ \begin{matrix} d^n \\ \dot{d}^n \Delta t \\ \ddot{d}_s^n \Delta t^2 \\ d_f^n \Delta t^2 \\ \mathcal{R}^n \Delta t^2 \end{matrix} \right\} \tag{28}$$

where $\mathcal{A}_{\text{strong}}$ is a five-dimensional amplification matrix. The coefficients of $\mathcal{A}_{\text{strong}}$ are dependent on the mass, damping and stiffness matrices, as well as on the time integration parameters $\rho_\infty^s$ and $\Delta t$.

The system (26) is unconditionally stable if the spectral radius $\rho_\infty \leq 1$ for any time step size $\Delta t$. According to (Dettmer and Perić, 2012), the spectral radius of an amplification matrix of dimension $\kappa$ is defined as

$$\rho_\infty = \max(|\lambda_1|, |\lambda_2|, \cdots, |\lambda_\kappa|) \tag{29}$$

Here, $\lambda_i$ are the eigenvalues of the amplification matrix. The eigenvalues of $\mathcal{A}_{\text{strong}}$ were evaluated by Joosten et al. [43] for various parameters. Their evaluations suggested that unconditional stability could be achieved in all cases if the spectral radius $\rho_\infty \leq 1$. Thus, in this paper, we specify $\rho_\infty^s = 0.5$ for elastic bodies (Dettmer and Perić, 2006).

The partitioned coupling scheme considered in the present work is based on the force predictor for the solid problem [25,34]. The coupling formulation is Dirichlet–Neumann coupling, in which the interface traction from the fluid solver is considered as a Neumann boundary condition for the solid domain, and the interface velocity is applied as a Dirichlet boundary condition for the fluid problem. The procedure of the scheme is presented in Algorithm 1.

---

**Algorithm 1.** Partitioned coupling scheme.

---

1. Perform the traction force predictor for interface traction pressure $P_s^n$ and velocity $u_s^n$.
2. Each FSI iteration between $t \in [t^n, t^{n+1}]$:
    (a) Solve structural Eq. (15) using known interface traction $P_s^n$;
    (b) Apply Dirichlet velocity continuity condition
$u_f^{n+1} = u_s^{n+1}$ on interface $\Gamma$;
(c) Set $\dot{d}_f^{n+1} = u_f^{n+1}$ and update the fluid dynamic mesh;
(d) Solve ALE fluid Equation (10) for new interface traction $P_s^{n+1}$ and apply Neumann condition
$P_s^{n+1} = P_f^{n+1}$ on interface $\Gamma$.

---

In Algorithm 1, a structural solver is employed to accept external fluid forces. A fluid solver is used to accept the prescribed motion of the interface boundary. The structural and fluid solvers are used to proceed in a predictor–corrector format by constructing an iterative interface force correction and the force transfer at each sub-iteration. The schematic for the partitioned iterative coupling of the ALE fluid solver with the structural solver is depicted in Figure 1.

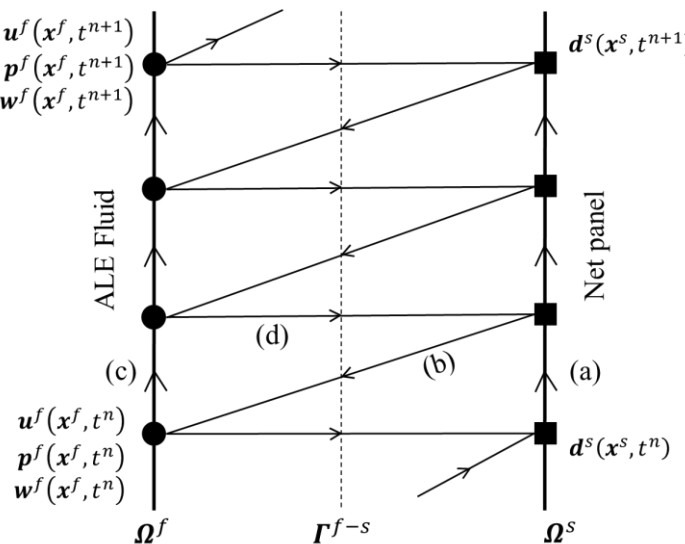

**Figure 1.** Schematic of predictor–corrector procedure for ALE fluid and net panel coupling: (**a**) Solve structural displacement; (**b**) transfer predicted structural displacement, where $u^f = u^s = w^f$ and update the fluid dynamic mesh; (**c**) solve ALE fluid equations and (**d**) forces corrections.

It is worth mentioning that the partitioned coupling scheme in this paper has second-order accuracy in time. Therefore, the time step required to obtain accurate numerical solutions is tolerably large. Moreover, the spectral ratio $\rho_\infty \leq 1$ to meet the partitioned scheme's unconditional stability. Alongside the stability to provide the partitioned fluid–structure coupling, the force correction is much more compatible with strong partitioned coupling than the displacement correction in a predictor–corrector framework.

## 5. Numerical Setup and Verification

### 5.1. The Net Panel

Two knotless square-mesh flexible net panels are examined in this study. The dimension of 0.25 m × 0.25 m is applied to both of them. The nets are made of polyethylene terephthalate (PET) and this material is commonly used for offshore aquaculture cages. The roughness of the net twine could be overlooked by using this material; thus, the net twines are considered as smooth cylinders in this study. A physical model of its geometry and physical properties, which have been used for numerical simulations, is displayed in Figure 2.

The parameters of the net twines and solidities are illustrated in Table 1. The velocity of the steady flow ranges from 0.15 m/s to 0.75 m/s with an increment of 0.15 m/s. Meaning that the effect of turbulence is not considered in this study. The corresponding Reynolds numbers are from 300 to 1500 based on the net twine diameter and the flow velocity. The drag coefficient is defined as $C_D = F_D / \left(0.5\rho^f U_\infty^2 A_p\right)$, where $F_D$ is the time-averaged drag force estimated from the numerical simulations, $\rho^f$ is the fluid density, and $A_p$ is the projected area of the flexible net panel. During the simulation, the top of the net is connected to a rigid rod which remains completely fixed. The rest of the net panel is set free and with no sinkers attached in the bottom, so that the effect of the sinkers can be eliminated.

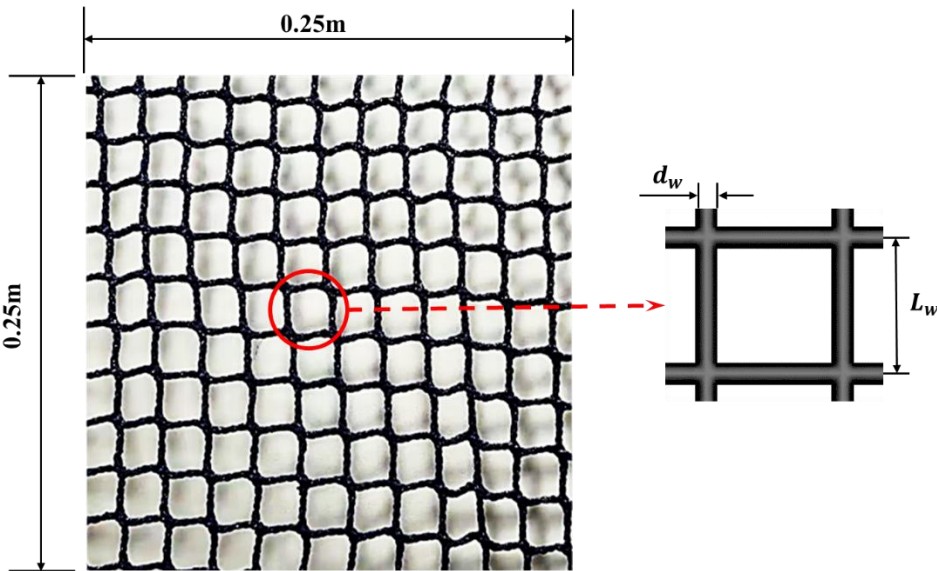

**Figure 2.** A physical model of the net panel and its geometrical description.

**Table 1.** Parameters of the net panels. ($S_n = \frac{2d_w}{l_w}$, and $d_w$ and $l_w$ are the diameter and length of the mesh bar [44]).

| $S_n$ | $d_w$ (mm) | $l_w$ (mm) |
|---|---|---|
| 0.16 | 2 | 25 |
| 0.33 | 2 | 12 |

The density of the net is 1375 kg/m³ and the modulus of elasticity, $E = 2.95 \times 10^9$ Pa. With these material properties, the net panels in the present study are much heavier and much stiffer than that used in the previous experimental studies [15,45,46].

### 5.2. Computational Domain and Boundary Conditions

The computational domain in this work uses the symmetry of the problem and is set at 1.5 m (length) × 0.5 m (width) × 0.5 m (height) for the model, as shown in Figure 3. The domain parameters are larger than those used by Bi et al. [15] for the effect of the computational domain on the wake behind the nets. Considering the fluid–solid interface, the tetrahedral grids are used to mesh the domain separately. Considering this, Dirichlet–Neumann coupling can be achieved via the interface between the fluid and the solid domains. Depending on the Reynolds number, grids near the net twines are set from $7 \times 10^{-5}$ m to $2.5 \times 10^{-4}$ m to capture the flow transition. The total number of grids is approximately $1.2 \times 10^7$.

As addressed in Section 3.1, a uniform streamwise velocity (i.e., $U_\infty$) is imposed at the upstream inlet boundary, while a pressure outlet boundary condition is applied at the downstream outlet boundary. The lateral sides of the computational domain are set as slip boundaries using symmetry conditions. In particular, the interface traction from the fluid solver is considered as a Neumann boundary condition for the solid domain, and the interface velocity is applied as a Dirichlet boundary condition for the fluid problem.

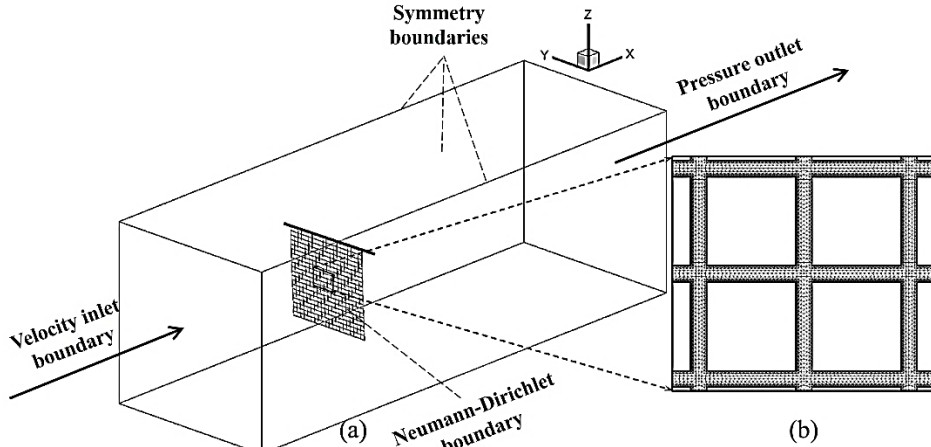

**Figure 3.** The schematics of the computational domain: (**a**) overall fluid domain and the solid subdomain, and (**b**) grids of the undeformed net panel in the solid subdomain.

### 5.3. Numerical Verification

To verify the present numerical method, the drag coefficients of the present numerical simulations are compared with analytical models that consider the solidity ratio $S_n$, the attack angle $\alpha$, and the Reynolds number $Re$. Since this paper studies flexible net panels, the net panel's attack angle should be considered. Aside from the angle of attack, the effects of the solidity ratio and the Reynolds number are too profound to be negligible. Two analytical models used for estimating the drag coefficients of flexible net panels are illustrated in Table 2.

**Table 2.** Analytical models for estimating the drag coefficients.

| Researchers | Empirical Formulae |
|---|---|
| Aarsnes et al. [47] | $C_D = 0.04 + \left(-0.04 + S_n - 1.24 S_n{}^2 + 13.7 S_n{}^3\right) \sin \alpha$ |
| Kristiansen and Faltinsen [19] | $C_D = C_D^{cyl} \dfrac{S_n(2-S_n)}{2(1-S_n)^2}$ |

A seventh-order polynomial is employed to approximate the drag coefficient of a circular cylinder (i.e., single net twine) at $10^{3/2} \leq Re \leq 10^4$, as can be seen in Kristiansen and Faltinsen [19]. This formula is valid for a knotless fabric net model with $S_n \leq 0.5$ and an angle of attack in the range of $\pi/4 \leq \alpha \leq \pi/2$. The corrected towing velocity of the net twines, which considers relative motion, gives more accurate results. Figure 4 exhibits the dependence of the drag coefficients of the flexible net panels on the Reynolds number. The drag coefficients of the present numerical simulations and the analytical model predictions are illustrated in Table 3.

The overall trends of the drag coefficients in Figure 4 decrease as the Reynolds number increases. When $Sn = 0.16$, the present numerical simulations overestimate the drag coefficients comparing with those of Aarsnes et al. [47]. The present numerical simulations underestimate the drag coefficients compared to analytical predictions by Kristiansen and Faltinsen [19]. When $Sn = 0.33$ and the Reynolds number is less than 600, the present numerical simulations slightly overestimate the drag coefficient when compared with those of the analytical predictions. However, when the Reynolds number is over 600, the present numerical simulations underestimate the drag coefficients compared to the analytical predictions.

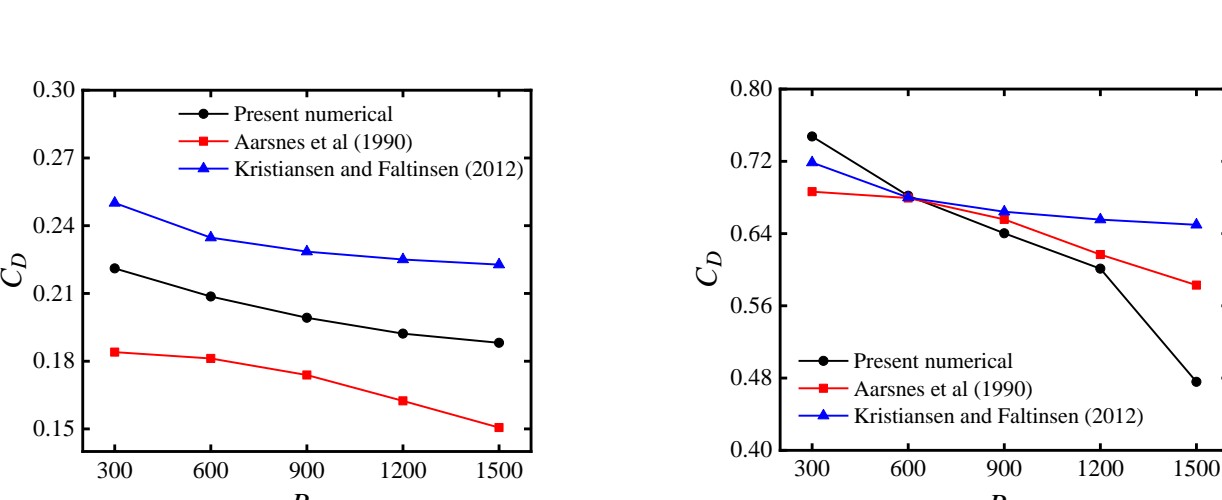

**Figure 4.** Dependence of the drag coefficients of the flexible net panels on the Reynolds number [19,47].

**Table 3.** Comparison of numerical predictions and analytical models of drag coefficients.

| *Re* | Drag Coefficient/Relative Difference | | | | | |
|---|---|---|---|---|---|---|
| | Num. | Num. | Aarsnes et al. [47] | | Kristiansen and Faltinsen [19] | |
| | *Sn* = 0.16 | *Sn* = 0.33 | *Sn* = 0.16 | *Sn* = 0.33 | *Sn* = 0.16 | *Sn* = 0.33 |
| 300 | 0.221 | 0.747 | 0.184 −16.7% | 0.686 −8.2% | 0.250 13.1% | 0.719 3.7% |
| 600 | 0.209 | 0.682 | 0.181 −13.4% | 0.679 −0.4% | 0.235 12.4% | 0.680 0.2% |
| 900 | 0.199 | 0.640 | 0.174 −12.6% | 0.656 2.5% | 0.229 15.1% | 0.664 3.8% |
| 1200 | 0.192 | 0.601 | 0.162 −15.6% | 0.617 2.7% | 0.225 17.2% | 0.655 8.9% |
| 1500 | 0.188 | 0.476 | 0.151 −19.7% | 0.583 22.5% | 0.223 18.6% | 0.650 36.5% |

Table 3 shows that when $Sn = 0.16$, the present numerical simulations overestimate the drag coefficients by approximately 15.6% on average compared to those of Aarsnes et al. [47]. Compared to the analytical predictions from Kristiansen and Faltinsen [19], the present numerical simulations, on average, underestimate the drag coefficients by roughly 15.3%. Xu and Qin [1] reported that the analytical model in Aarsnes et al. [47] did not consider the effect of the Reynolds number; this may cause the discrepancy between the analytical prediction and numerical simulations/experimental tests. The analytical models in Kristiansen and Faltinsen [19] consider the effect of the Reynolds number and the relative motions between the fluid and the netting, but it seems that the interference between net twines has been neglected. In particular, the net panel's flow field is so complicated that it should be considered to obtain more accurate results.

When $Sn = 0.33$, the relative difference in the drag coefficients between the present numerical simulations and the analytical predictions from Aarsnes et al. [47] is below 10%, under the condition that the Reynolds number is from 300 to 1200. The Reynolds numbers used in Aarsnes et al. [47] range from 200 to 1400; thus, further comparison between their predictions and the present numerical simulations seems erratic when $Re > 1400$. A perfect match between the present numerical simulations and the analytical predictions from Kristiansen and Faltinsen [19] has been achieved, except for $Re = 1500$. This implies that the present numerical model predicts the drag coefficient of a flexible net panel reasonably well when $Re < 1500$.

## 6. Results and Discussion

### 6.1. Velocity Distribution

The wake forms behind the net panel when the flow passes through it. Figures 5 and 6 depict the velocity distributions of two flexible net panels at different velocities. We see that the width of the wake is nearly the same as that of the net panel when the flow velocity is small. The width of the wake in the far field diminishes gradually as the flow velocity increases. This is mainly because the reconfiguration of the net causes a change in the pressure gradient, leading to the far-field wake diminishment. There is a slight velocity reduction upstream of the net panel; the velocity reduction in the downstream far-field region is relatively large compared to the upstream velocity reduction. The flow velocity reaches the steady state in the far-field region. However, the velocity accelerates dramatically in the near-field region where the wake flow does not block flow the net.

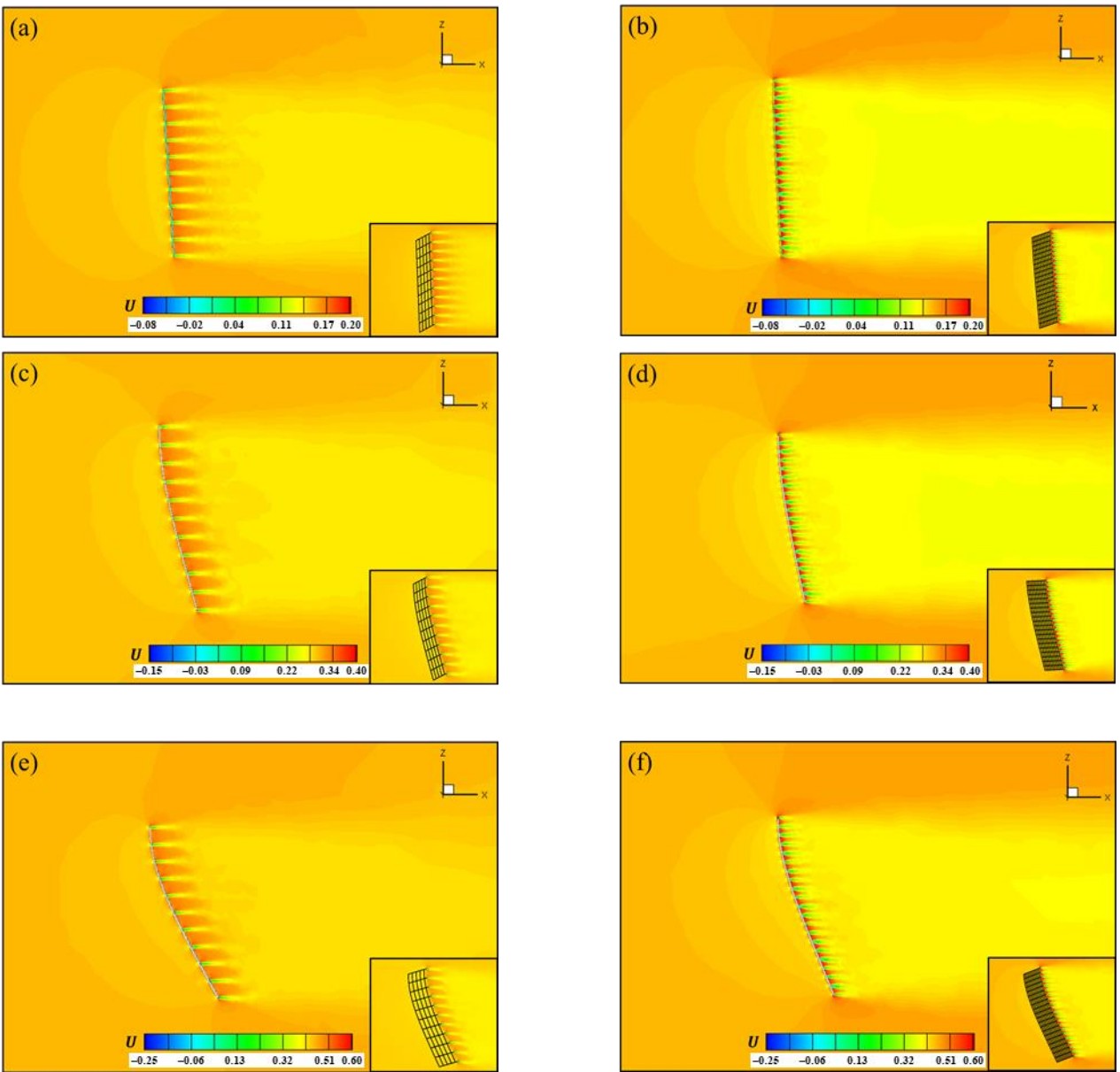

**Figure 5.** Velocity distributions of the flow at the angles of attack between $\pi/3$ and $\pi/2$ for net solidities $Sn = 0.16$ (**left** column) and $Sn = 0.33$ (**right** column): (**a,b**) $U_\infty = 0.15$ m/s; (**c,d**) $U_\infty = 0.30$ m/s; (**e,f**) $U_\infty = 0.45$ m/s.

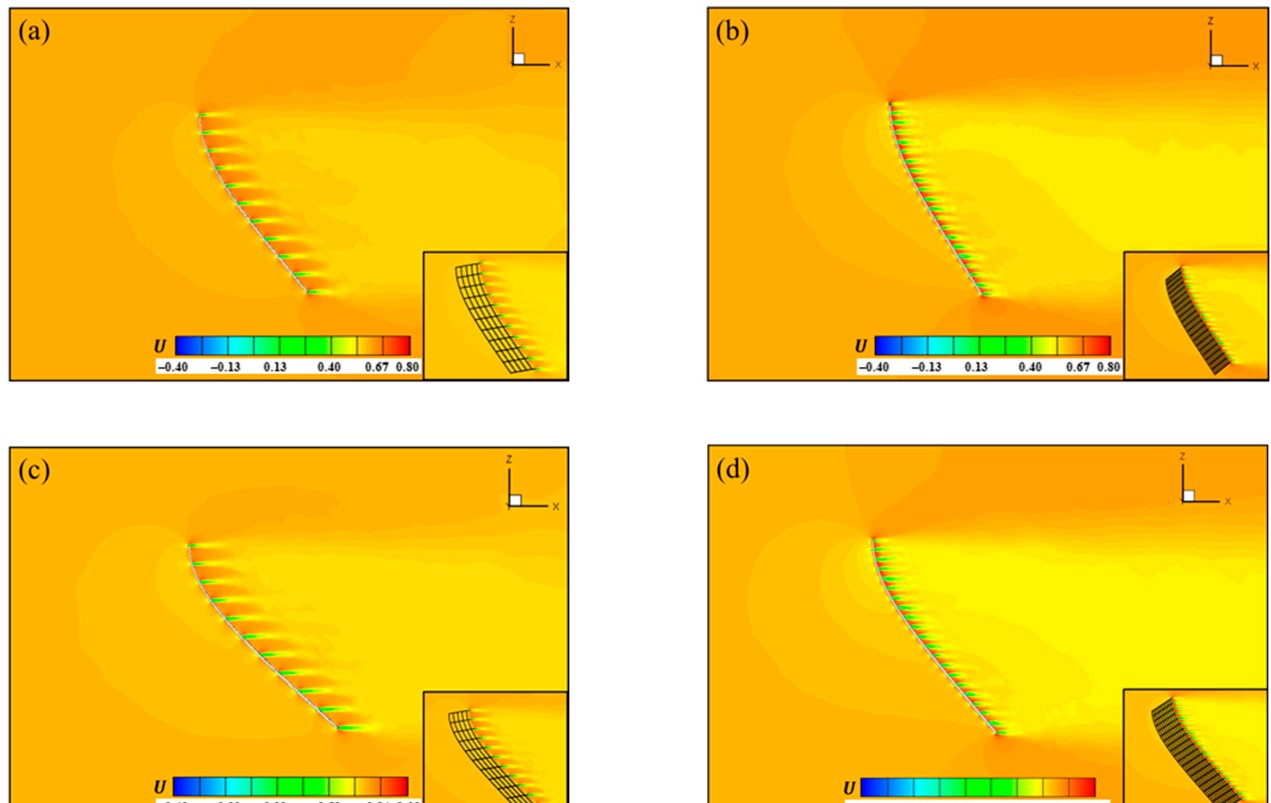

**Figure 6.** Velocity distributions of the flow at the angles of attack between $\pi/4$ and $\pi/3$ for net solidities $Sn = 0.16$ (**left** column) and $Sn = 0.33$ (**right** column): (**a**,**b**) $U_\infty = 0.60$ m/s; (**c**,**d**) $U_\infty = 0.75$ m/s.

Figures 5 and 6 present distinct flow patterns where the velocity reduction increases alongside the net solidity, meaning that higher solidity can cause a greater shielding effect in the flow field. As the net solidity reaches 0.33, we see clear interference between net twines. This interference causes energy losses when the flow passes the net twines and further affects the flow transition. Consequently, higher solidity results in greater velocity reduction. There is no such interference between net twines when $Sn = 0.16$. The velocity reduction is relatively large if the free stream flow velocity is small; this could be attributed to the large drag coefficient at a lower Reynolds number [48].

The velocity reduction becomes smaller as the inclination angle of the net increases. The inclination angle of the net is denoted as $\pi/2 - \alpha$, where $\alpha$ is the angle of attack. In other words, the velocity reduction decreases as the attack angle decreases. This is consistent with the numerical studies in Zhao et al. [13,14]. Moreover, a large inclination angle produces a distinct velocity gradient that could cause severe fluctuations regarding velocity reduction.

It appears that the flow characteristics of the nets may not only depend on the solidity ratio and the angle of attack but also on the flow regime (i.e., *Re*). Hence, two flow regimes are identified in this paper according to Zdravkovich [49]:

(1) Transition-in-wake state of flow (**TrW**).
(2) Transition-in-shear-layers state of flow (**TrSL**).

In the **TrW** state, the laminar periodic wake becomes unsteady at a higher Reynolds number further downstream. The transition spreads from upstream alongside the Reynolds number, until the eddy becomes turbulent during its formation. This **TrW** state can be divided into two phases, and they are defined as **TrW₁** and **TrW₂**. In the second phase **TrW₂**, the irregular eddy transition happens during its formation, where the Reynolds number, $Re \in [220 \sim 250, 350 \sim 400]$.

In the **TrSL** state, or subcritical state, the second transition occurs along the free shear layers while the boundary layers remain fully laminar. There are three phases of transition along the free shear layers, namely **TrSL₁**, **TrSL₂** and **TrSL₃**. In the first phase **TrSL₁**, the transition eddy formation happens, where the Reynolds number $Re \in [350 \sim 400, \ 1000 \sim 2000]$. Depending on the $Re$, this paper examines the interactions between the fluid and the flexible net panels within the **TrW₂** and **TrSL₁** regimes.

In the **TrW₂** regime, the transition of the irregular eddy happens during its formation. These irregular eddies facilitate the velocity reduction in the wake, causing greater velocity reduction at lower Reynolds numbers (e.g., $Re < 600$). The loss of velocity in the wake occurs due to a loss of momentum [48]. In the **TrSL₁** regime, the flow in the transition zone is free shear flow and the boundary layers remain laminar. This produces a moderate velocity reduction as the Reynolds number rises. Bi et al. [50] also presented the velocity distribution across the wake to the outer region, showing experimental evidence of how the velocity was distributed in the transition zone.

Figure 7 exhibits the magnitude of the flow velocity $u$ along the centerline of the net panel at different velocities. The overall trends of these data demonstrate that the velocity reduction exists due to the net panel's shielding effect, and that the velocity reduction increases alongside the net solidity. The average velocity reductions for these two nets are 7.5% and 12% with regard to $Sn = 0.16$ and $Sn = 0.33$, respectively. As the attack angle decreases, the velocity reduction for $Sn = 0.16$ drops from 9% to 7%. In comparison, these values for $Sn = 0.33$ slide from 15% to 10%.

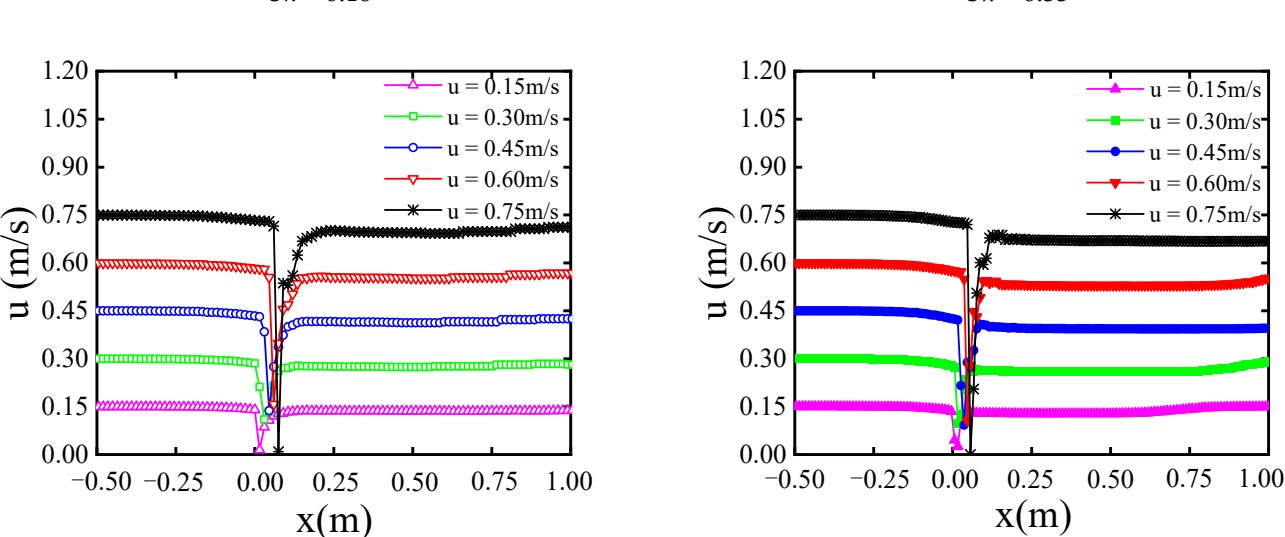

**Figure 7.** The magnitude of the flow velocity $u$ along the centerline of the net panel at different velocities (the 0.0 m along the x-axis denotes the center-line of the net).

To estimate the velocity reduction caused by the net's shielding effect, Løland [48] proposed a linear correlation between the velocity reduction factor and the drag coefficient. That is, $r = 1 - 0.46 C_D$, where $r$ is the velocity reduction factor. This linear correlation indicates that the velocity reduction decreases linearly with the drag coefficient. In our simulations, the trends remain the same when $Sn = 0.16$, and the linear correlation is $r = 1 - 0.77 C_D$. However, the correlation becomes $r = 0.688 + 0.386 C_D$ when $Sn = 0.33$. This is because Løland [48] does not consider the interactions of the net twines, which could affect the drag coefficient of the net and the velocity distribution.

Aside from the linear correlation, several values for the net panel/cage's velocity reductions are summarized by Xu and Qin [1]. Most of them favor the value of 10% [12–15,45,51]. It is better to compare these values in specific conditions regarding the net solidity, attack angle and flow regimes (i.e., $Re$). Moreover, to avoid underestimating the velocity reduction, a value of 20% has been recommended [1].

The origin drifts along the *x*-direction due to the deformation of the net (see Figure 7). The offset increases alongside the flow velocity; however, a slight difference exists between these two nets with different solidities. The net with higher solidity shows less offset because of the net panel's increased shielding effect and weight. The largest offset reaches 0.078 m when the flow velocity is 0.75 m/s and $Sn = 0.16$.

Figure 8 presents the magnitude of the flow velocity *u* along *y*-direction at different distances downstream of the net panel center. The velocity exhibits a U-shaped distribution in the overall range of $-0.125$ m $\leq y \leq 0.125$ m. However, the velocity along the *y*-direction at different angles of attack in Zhao et al. [13] is more likely a V-shaped distribution (see Figure 16 in Zhao et al. [13]). We note that the net panel being studied in Zhao et al. [13] does not consider the deformation; instead, by directly assigning different inclination angles, the authors examined the velocity distribution of an undeformed net panel. The deformation could be the cause of the differences in velocity distribution between the present study and Zhao et al. [13].

From Figure 8, we see distinct velocity reductions concerning different net solidities. Higher solidity causes greater velocity reduction, mainly because of the increased shielding effect. However, the relative difference in velocity reduction between these two net panels drops from 40% to 30% when flow velocities increase from 0.15 m/s to 0.75 m/s. This means that the velocity reduction differences between these two net panels become smaller as the flow velocity increases, because the shielding effect becomes weak when the inclination angle grows. Under this circumstance, the flexible nets choose to "go with the flow" to reduce the flow impact and reduce drag forces.

$X = 0.13$ m $\qquad\qquad\qquad\qquad$ $X = 0.25$ m $\qquad\qquad\qquad\qquad$ $X = 0.50$ m

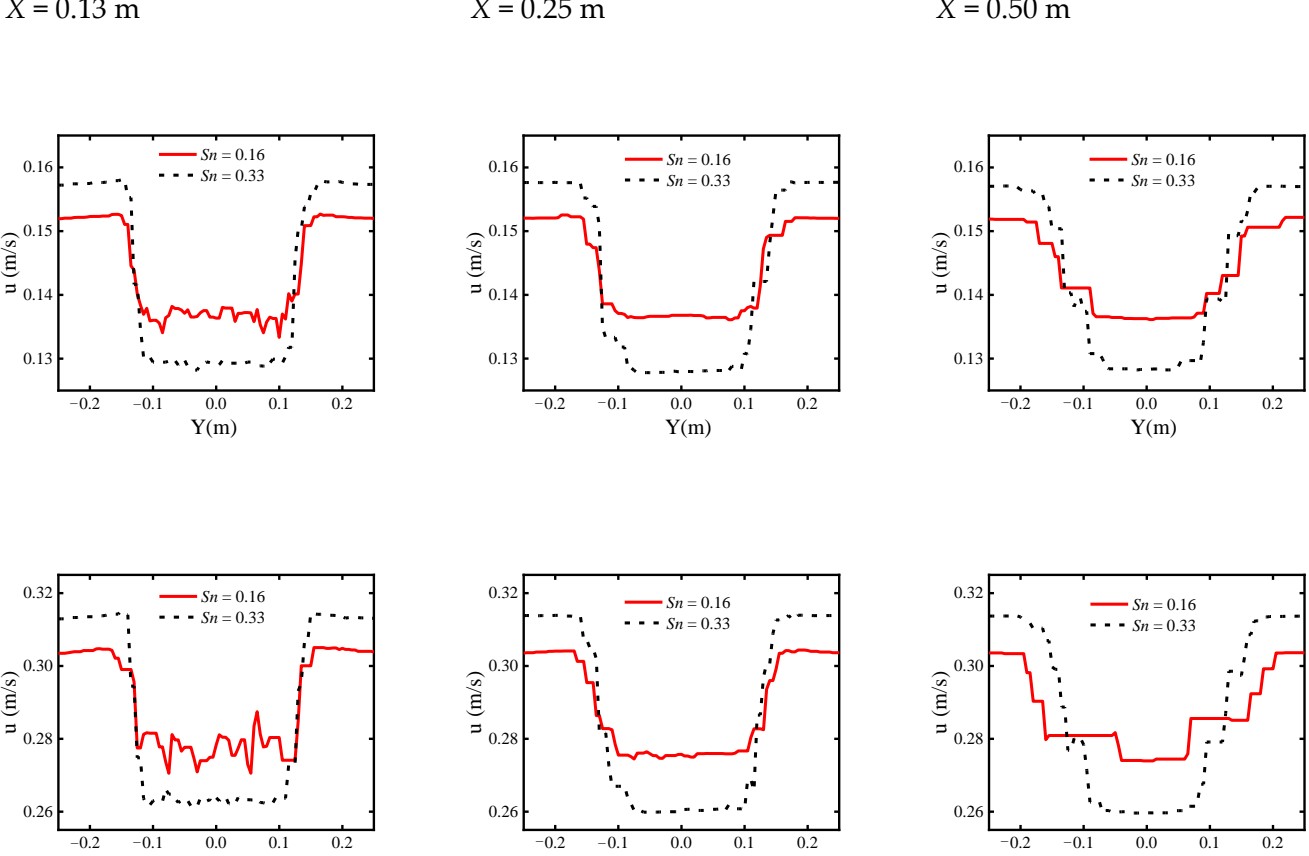

**Figure 8.** *Cont.*

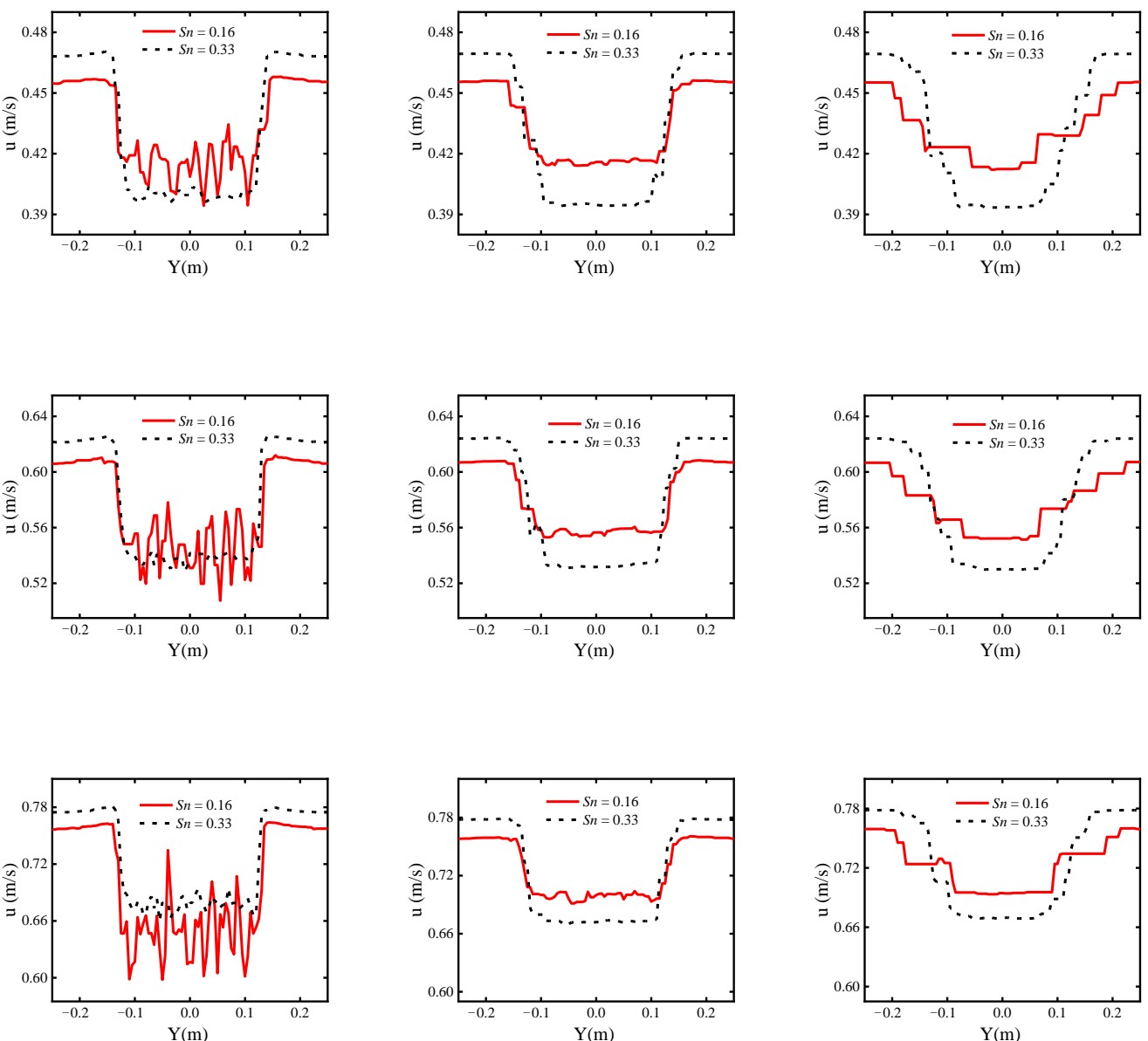

**Figure 8.** The magnitude of the flow velocity *u* along *y*-direction at different distances downstream of the net panel center. Here, the flow velocities from top to bottom are from 0.15 m/s to 0.75 m/s with an increment of 0.15 m/s.

We separately define the distances $X = 0.13$ m and $X = 0.50$ m as near-field and far-field, while $X = 0.25$ m is a distance in the middle of the near field and far field. The velocity reduction is larger in the near field than those beyond it. It appears that the velocity distribution in the middle of near field and far field is more stable than those in the near field and far field. Severe velocity fluctuations exist in the near field when $Sn = 0.16$. There is no interference between net twines at such a low solidity; however, the large opening in the net allows the flow to develop aggressively. Considering this, with blocking from the net twines, the severe velocity fluctuations appear in the near field. Furthermore, the velocity fluctuations become much more severe when the flow velocity rises to 0.6–0.75 m/s. There are also some obvious velocity fluctuations at a solidity of 0.33 if the flow velocity is over 0.6 m/s. Compared to the near-field velocity distribution, far-field velocity exhibits a zigzagged distribution, mostly because there are velocity gradients in the far field induced by the net deformation.

### 6.2. Recirculation and Vortex Formation

Recirculation is a specific condition alongside flow separation from a bluff body. The flow separation immediately generates a low-pressure area that rolls the flow back into the body, resulting in a circulating vortex or pair of vortices. Therefore, it is necessary to quantify the recirculation and its size (e.g., width), given their close connections with the body pressure and the drag acting on the net panel. The recirculation creates remarkable variations in drag compared to those situations in which the flow remains attached to the surface of the body. Three typical sections on the net panels, namely the top, middle and bottom (Figure 9), have been selected to monitor the recirculation and vortex formation behind the vertical net twines. There is no need to monitor the horizontal net twines since they are always parallel to the flow direction.

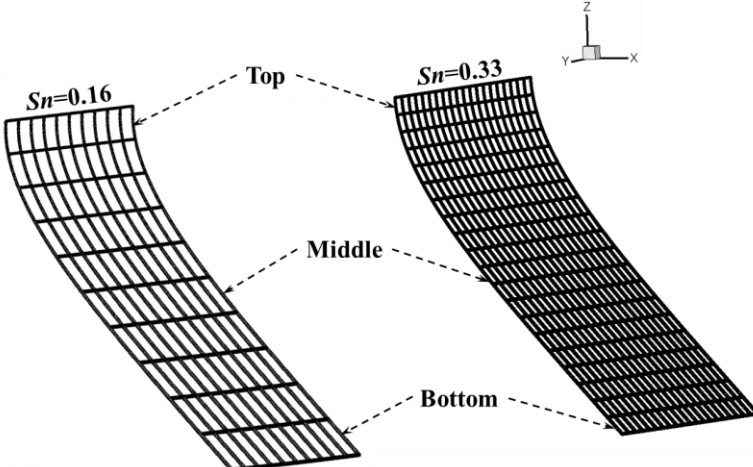

**Figure 9.** The selected sections on the net panels are used to monitor vortex formation behind vertical net twines. Here, these two deformed net panels are under a flow velocity of 0.75 m/s.

Figure 10 depicts two-dimensional instantaneous streamlines of three selected sections at different Reynolds numbers in the same time frame. The vortices generated by flow separation and recirculation exhibit distinct characteristics. These are: (1) a single vortex appears behind the net twine when the solidity $Sn = 0.16$, whereas when $Sn = 0.33$, there is a pair of vortices behind the net twine; (2) vortices distribute both in the shear layers and the wake when $Sn = 0.16$, but when $Sn = 0.33$, vortices distribute mostly in the wake rather than in the shear layers; (3) vortices decay faster in the top section than those in the middle and bottom sections; (4) when the flow transfers from $TrW_2$ to $TrSL_1$, the recirculation width increases when $Sn = 0.16$, and it decreases when $Sn = 0.33$. This is related to the transition of the wake and shear layers; (5) When $Sn = 0.16$, the reattachment of the shear layers occurs in the $TrSL_1$ regime. The reattachment moves upwards from one side to the other of the net twine in the middle and bottom sections. There is no significant change for the attachment in the top section.

We would expect in-phase vortex structures since there is no mutual interference between net twines if the net solidity is 0.16. However, due to the net deformation, the single vortex exhibits a distinct formation even at the same frequency in different net sections. As the Reynolds number increases and the flow is in the $TrSL_1$ phase, vortices form in the shear layers. This could be the reason that the velocity fluctuation happens along the $y$-direction in the case that $Sn = 0.16$. The pair of vortices behind the net twines for $Sn = 0.33$ are more likely wake interactions. This finding differs from the predictions in Løland [48] because the author assumes that the interactions between the net twines can be overlooked if the net solidity is less than 0.33. In fact, the distance between net twines significantly affects the shear layers at $Sn = 0.33$, causing the shear layers to be more unstable.

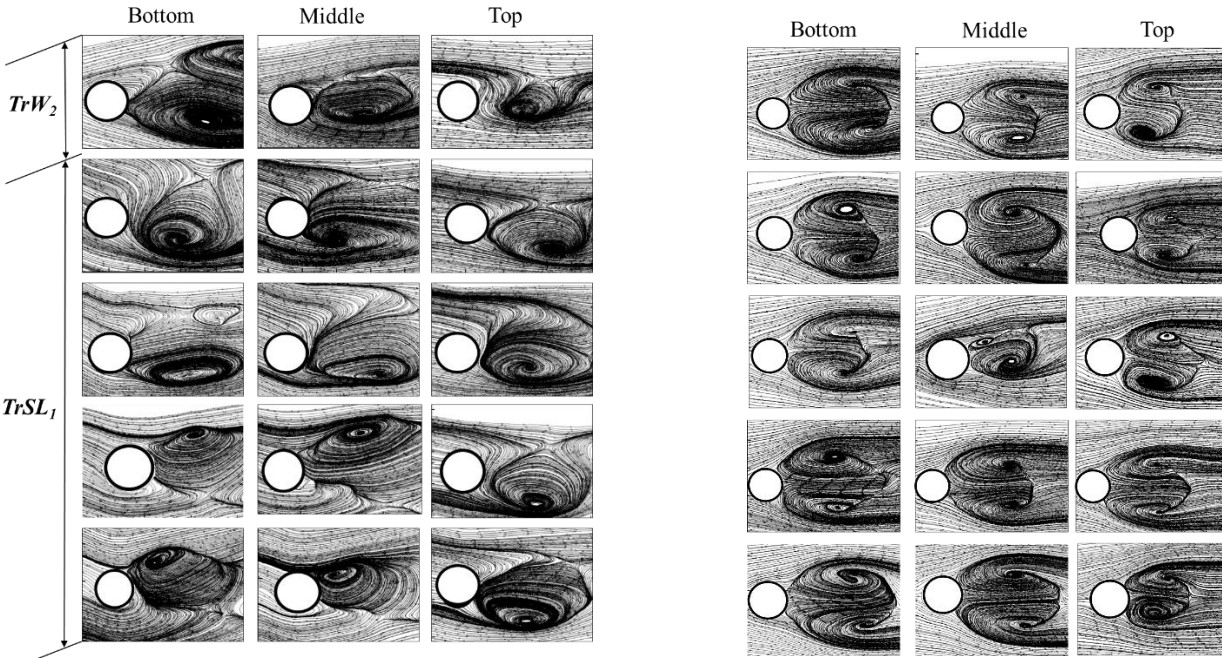

**Figure 10.** Two-dimensional instantaneous streamlines of three selected sections at different Reynolds numbers in the same time frame. (The top, middle and bottom are three typical sections on the net panels, as shown in Figure 9).

We note that vortices decay faster in the top section than those in the middle and bottom sections. This indicates that, as the inclination angle increases, the shielding effect becomes weak, resulting in less energy loss. Thus, the recirculation bubble size (width) in the middle/bottom section is greater than those in the top section, except for $Re = 1500$. The statistical evidence is shown in Figure 11. In the case that $Sn = 0.16$, when the flow switches from **TrW$_2$** to **TrSL$_1$**, the shear layers are supposed to reattach to the body surface, which they do, but this rarely happens in the top section of the net. The transition of the altered shear layers produces larger recirculation bubbles in the middle and bottom sections. Furthermore, the reattachment of the altered shear layers on the net twines should be responsible for the drag reduction. In the case that $Sn = 0.33$, where the shear layers have been disturbed, the shrink of the recirculation bubble in the top section is most likely attributed to the energy loss. However, this prediction needs a power spectral analysis and will not be addressed in the present study.

Figure 11 presents the dependence of the non-dimensional recirculation width ($W_r^*$) on $Re$. To compare the magnitudes of the recirculation size, we normalize the recirculation width with the diameter of the net twine $d_w$. Furthermore, the non-dimensional recirculation width ($W_r^*$) is measured from a single vortex since most pairs of vortices when $Sn = 0.33$ seem symmetrical.

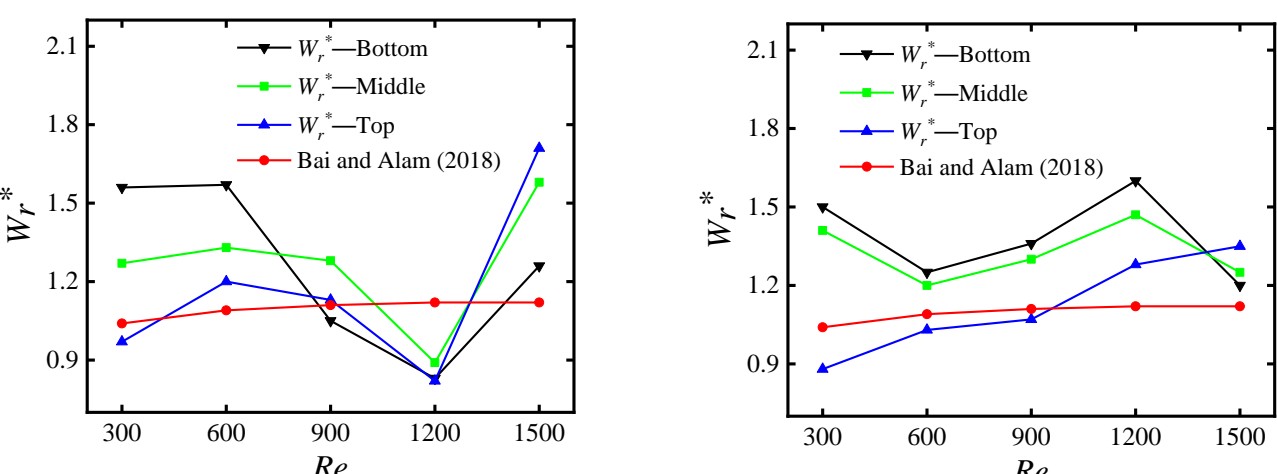

**Figure 11.** Dependence of the non-dimensional recirculation width ($W_r^*$) on the Reynolds number.

The overall trends of the non-dimensional recirculation width ($W_r^*$) are divided into two regimes, namely $TrW_2$ and $TrSL_1$. For $Sn = 0.16$ in the $TrW_2$ regime, $W_r^*$ increases alongside the Reynolds number, and a value around 1.57 is attained in the regimes where flow transfers from $TrW_2$ to $TrSL_1$. $W_r^*$ drops in the $TrSL_1$ regime where the minimum value is 0.83. However, in the post $TrSL_1$ regime, $W_r^*$ rises progressively to a maximum of 1.71 on the top section. This is mainly because the shear layers have been disturbed. For $Sn = 0.33$ in the $TrW_2$ regime, the value of $W_r^*$ decreases in the middle and bottom sections. However, it increases in the top section from the $TrW_2$ to the $TrSL_1$ regime and its value, in general, is smaller than those in the middle and bottom sections. The minimum $W_r^*$, approximately 0.88, is obtained at $Re = 300$. Additionally, in the middle and bottom sections, $W_r^*$ grows rapidly during the transition from the $TrW_2$ to the $TrSL_1$ regime and it reaches a peak value of 1.6. Then, $W_r^*$ falls in the post $TrSL_1$ regime.

The net panels could be considered as a combination of cylinders (i.e., net twines). It is well known that $W_r^*$ is related to the fluid force acting on a bluff body such as a cylinder, and that larger $W_r^*$ causes a lower pressure area behind the body. Moreover, the reverse flow in the wake of the body influences the streamwise vortices. According to Zdravkovich [49], larger $W_r^*$ can lead to a higher $C_D$ for circular cylinders. However, the net panel's flow characteristics are much more complicated than that of a system of cylinders. From this point of view, it may be helpful to compare the present simulation and the results from Bai and Alam [52], to facilitate a better understanding of how recirculation/vorticity behind net twines impacts flow characteristics of the net panel. Still, to provide more insights into the flow characteristics, studies on three-dimensional vortices of the net panel need to be conducted.

This paper uses the Q-criterion [53] to identify the iso-surfaces of the three-dimensional instantaneous vorticities. By decomposing the velocity gradient $\nabla \cdot U = S + \Omega$—where $S = \frac{1}{2}\left[\nabla \cdot U + (\nabla \cdot U)^T\right]$ is the rate-of-strain tensor and $\Omega = \frac{1}{2}\left[\nabla \cdot U - (\nabla \cdot U)^T\right]$ is the vorticity tensor—Q can be defined as $Q = \frac{1}{2}\left(|\Omega|^2 - |S|^2\right)$. Considering this, the iso-surfaces of the Q-criterion vorticities are exhibited in Figures 12 and 13.

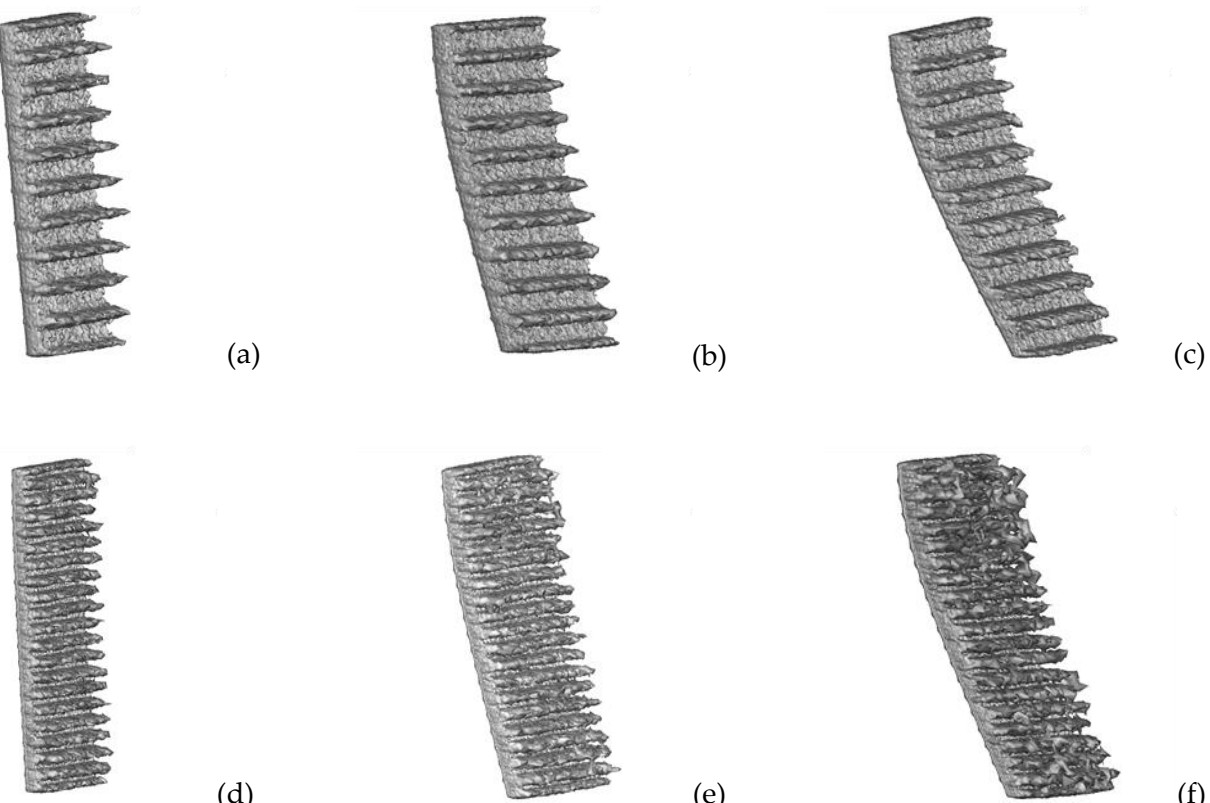

**Figure 12.** Three-dimensional instantaneous vorticity structures behind the flexible net panels at the angles of attack between $\pi/3$ and $\pi/2$: (**a**–**c**) $Sn = 0.16$; (**d**–**f**) $Sn = 0.33$.

The vortex shedding overall is well-organized but not as regular as the classical Karman vortex. The length of the vortex varies differently as the net deformation changes. It is easy to capture the vortex shedding in the horizontal net twines rather than in the vertical ones, meaning that the horizontal net twines have less impact on vortex formation. Meanwhile, vertical net twines significantly affect vortex formation, leading to delayed vortex shedding. These vortices for $Sn = 0.16$ (Figures 12a–c and 13a,b) have proven that mutual interference does not exist in this situation. However, from Figures 12d–f and 13c,d, we see remarkable wake interactions for $Sn = 0.33$. In particular, the wake interacts more severely at the angles of attack between $\pi/4$ and $\pi/3$, and the considered Reynolds number is above 1200. This indicates that, when the net solidity $Sn \geq 0.33$, there is significant interference between net twines that should not be neglected. Moreover, a higher-solidity net (i.e., $Sn = 0.33$) forms wake structures with more elongated shear layers.

Figure 13c,d show that vortices in the top sections of the net panel develop faster than those in the bottom sections, especially when the Reynolds number is higher than 1200. This is because, aside from the shielding effect, as the Reynolds number increases, the separated shear layers in top sections become more unstable. The transition of the shear layers and shear-layer vortices produces increased near-wake vortices, which leads to the breakdown of the vortices in both the shear layers and the near wake. Compared to vortices in the middle and bottom sections of the net panel, the energy loss induced by vortex interactions generates smaller vortices in the top sections. Furthermore, when $Sn = 0.33$, the shear layer attachment seems to be suppressed due to the vortex interactions. This is consistent with the two-dimensional demonstrations in Figure 10. These unstable shear layers and the delay of vortex shedding also contribute to the drag reduction.

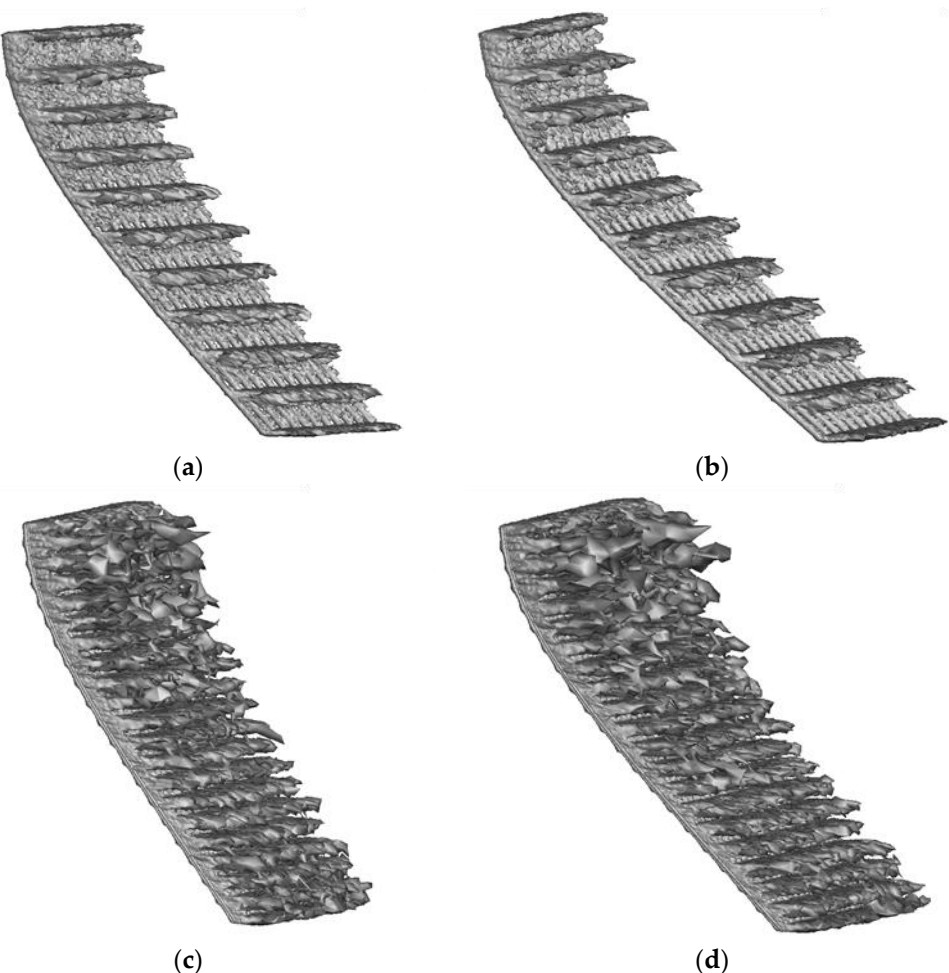

**Figure 13.** Three-dimensional instantaneous vorticity structures behind the flexible net panels at angles of attack between $\pi/4$ and $\pi/3$: (**a**,**b**) $Sn = 0.16$; (**c**,**d**) $Sn = 0.33$.

## 7. Conclusions

This paper proposes a coupled fluid–structure partitioned scheme to study the FSI of flexible net panels under steady flow. Using the FVM and FEM schemes, the coupled fluid–structure equations are solved separately in the fluid and solid subdomains. An arbitrary Lagrangian–Eulerian formulation is employed to accommodate the independent mesh motion at each time step. Then, the Dirichlet boundary conditions for the fluid and Neumann boundary conditions for the structure are imposed along the interface. Considering this, the variables, namely the forces, are transferred along the interface via a predictor–corrector scheme.

Two net panels with different solidities are examined based on the present coupled fluid–structure partitioned scheme. The results show that both the drag coefficient and the velocity reduction increase alongside the net solidity, but they decrease as the Reynolds number/attack angle increases. When the solidity $S_n$ equals 0.16, the relative difference in the drag coefficient between the present numerical simulations and analytical predictions is approximately 15%. When the solidity $S_n$ is 0.33, compared to the validated analytical model, the average difference is about 4.2% at $Re < 1500$. Hence, the present numerical model predicts the drag coefficient reasonably well. Greater velocity reduction exists at $Re < 600$, while moderate velocity reduction appears at $600 < Re < 1500$. The average values for these two nets are 7.5% and 12% regarding $Sn = 0.16$ and $Sn = 0.33$. The velocity distributions along the $y$-direction at different distances show distinct patterns: those in the middle of near field and far field are more stable. Severe velocity fluctuations have been observed in the near field when $Sn = 0.16$. This could be caused by the instability of the

free shear layers. Moreover, in terms of the near-wake vortex distribution, the low-solidity net favors forming vortices in the shear layers and the wake, while the high-solidity net prefers the wake rather than the shear layers. The non-dimensional recirculation width ($W_r^*$) is dependent on the solidity, the Reynolds number, and the angle of attack. When the net solidity $Sn \geq 0.33$, the flow shows that the wake interactions and the shear layers have been disturbed. However, the wake structures behind the net panel have more elongated shear layers, and vertical net twines have a predominant effect on vortex formation.

**Author Contributions:** Conceptualization, L.X. and Z.X.; methodology, L.X. and Z.X.; software, H.Q. and P.L.; validation, L.X., H.Q. and P.L.; formal analysis, Z.X.; investigation, L.X. and Z.X.; resources, P.L.; data curation, H.Q.; writing—original draft preparation, L.X., P.L. and Z.X.; writing—review and editing, L.X., H.Q. and Z.X.; visualization, H.Q. and P.L.; supervision, H.Q. and P.L.; project administration, H.Q.; funding acquisition, P.L. All authors have read and agreed to the published version of the manuscript.

**Funding:** This research was funded by the National Natural Science Foundation of China (Grant No. 51909040), the Natural Science Foundation of Heilongjiang Province (Grant No. LH2020E073), and the Key Technology Research and Development Program of Shandong (Grant No. 2020CXGC010702).

**Institutional Review Board Statement:** Not applicable.

**Informed Consent Statement:** Not applicable.

**Data Availability Statement:** Data available in a publicly accessible repository.

**Conflicts of Interest:** The authors declare no conflict of interest.

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
