# Peer review of "Numerical Modeling of Flexible Net Panels under Steady Flow Using a Coupled Fluid–Structure Partitioned Scheme"

_applsci, doi:10.3390/app12073399_

Round 1

Reviewer 1 Report

Comments:

In this paper, the flow characteristics of flexible net panels with large deformation was studied numerically, and a partitioned coupling scheme was proposed. The topic is interesting, and it is useful for the scientific and technical community. Therefore, I think this paper could be publishable if a few questions could be answered. Please see my comments below:

Comments:

  1. It would be appreciated if the authors could provide the turbulence model used in this paper. Because I see the governing equation of the fluid but do not see the turbulence model.
  2. What makes the authors state that the net panels in the present study are much stiffer than that sued in the previous experimental studies? What are the material properties for the previous studies?
  3. What is the y plus used in the present study?
  4. What causes the difference of the recirculation width between the net and the cylinder in Fig.11?

Reviewer 2 Report

The manuscript presents a coupling methodology in order to accurately assess the fluid structure interaction phenomena around a flexible net panel. The influence of the net angle of attack, solidity ratio and the flow regime on the velocity flow field is investigated.

The English is decent and the manuscript follows a logical structure.
The presented governing equations for the fluid, structure and the FSI interface are correct. The methodology is sound as well as the TrW2/TrSL1 analysis.

The reviewer has some minor comments:

It should be specified in the abstract that Sn denotes the solidity ratio.
The introduction section is succinct and a throughout review is given, however, the nomenclature of the three methodologies for net numerical analysis is a bit confusing since generally, FEM could also a CFD method. 

Fig. 3 -- The reviewer would suggest adding boundary conditions into the figure to increase clarity of the numerical setup.

Fig. 7 -- The reviewer would suggest adding in the caption that the 0.0 m along the x-axis denotes the center-line of the net
